# Revisiting B2T: Discovering and Mitigating Visual Biases through Keyword Explanations

**Aïda Asma***
*University of Amsterdam*
*Aidaasma.aa@gmail.com*

**Faissal El Kayouhi***
*University of Amsterdam*
*Faissalkay@gmail.com*

**Joey Laarhoven***
*University of Amsterdam*
*Joeylaarhoven@gmail.com*

**Fiona Nagelhout***
*University of Amsterdam*
*Fiona.nagelhout@gmail.com*

**Reviewed on OpenReview:** *https://openreview.net/forum?id=5GS1q65pv6*

## Abstract

This work aims to reproduce and extend the findings of "Discovering and Mitigating Visual Biases through Keyword Explanation" by Kim et al. (2024). The paper proposes the B2T framework, which detects and mitigates visual biases by extracting keywords from generated captions. By identifying biases in datasets, B2T contributes to the prevention of discriminatory behavior in vision-language models. We aim to investigate the five key claims from the original paper, namely that B2T (i) is able to identify whether a word represents a bias, (ii) can extract these keywords from captions of mispredicted images, (iii) outperforms other bias discovery models, (iv) can improve CLIP zero-shot prompting with the discovered keywords, and (v) identifies labeling errors in a dataset. To reproduce their results, we use the publicly available codebase and our re-implementations. Our findings confirm the first three claims and partially validate the fourth. We reject the fifth claim, due to the failure to identify pertinent labeling errors. Finally, we enhance the original work by optimizing the efficiency of the implementation, and assessing the generalizability of B2T on a new dataset.

## 1 Introduction

Datasets are a fundamental part of developing vision-language models, such as image classifiers. However, biased datasets can negatively impact performance (Torralba & Efros, 2011) and can cause the model to demonstrate discriminatory behavior towards certain groups (Mehrabi et al., 2022).

It can be difficult to detect and mitigate visual biases. Biases within images are harder to detect than biases within statistical data, as structural differences in data are less explicit in images, making them harder to define and extract. Previous work by Wiles et al. (2023) extracts captions from synthesized images. These descriptions provide detailed information about the presence of visual biases, but are too complicated for effective debiasing. Sagawa et al. (2020a) introduced Distributionally Robust Optimization (DRO), which minimizes the training loss over a set of pre-defined bias groups.

"Discovering and Mitigating Visual Biases through Keyword Explanation" by Kim et al. (2024) proposes a pipeline to detect and remove visual biases from datasets in an unsupervised manner. They introduce the Bias-to-Text (B2T) framework, which extracts common keywords from the captions of mispredicted images. These keywords are interpreted as biases and utilized to train the debiased classifier DRO.

---

*These authors contributed equally to this work

In this paper, we aim to reproduce the findings of Kim et al. (2024). The following contributions are made:

- The work of Kim et al. (2024) is reproduced to investigate the validity of their claims.

- The codebase delivered by the authors is incomplete, as it does not provide a way to obtain the data and code required to reproduce the different figures and tables. We contribute our re-implementation and documentation of the model performance and evaluation.

- We optimize the original pipeline to increase the computational speed and make more efficient usage of the GPU. Additionally, this reduces the environmental impact.

- We test the generalizability of the B2T framework on a new dataset.

## 2 Scope of reproducibility

Kim et al. (2024) focus on two types of biases. The first is spurious correlations. Here, the model learns patterns without semantically meaningful links to a given class, instead of the correct features (Simon, 1954). This leads to discriminatory behavior based on gender or race, for example. The second is distribution shifts, where the test data is drawn from a different distribution than training data (Recht et al., 2019). This difference in distribution causes the training performance of the model to not be applicable to future test scenarios, rendering the model useless in the worst case.

Kim et al. (2024) focus on identifying and mitigating these biases by employing an in-processing technique. Below, we outline the key claims of the original authors that will be addressed in this work. All models and algorithms mentioned here are discussed in section 3.

1. **CLIP score validation:** The CLIP score is a valid method to identify whether a single word or a set of words represents a bias or not. A word does not represent a bias when it does not negatively affect a classifier's performance, compared to a random keyword.

2. **Bias identification:** B2T can extract keywords from mispredictions that represent biases in datasets. Here, the authors make a distinction between detecting known and unknown biases. For example, there is a known gender bias in CelebA that B2T can correctly detect. Furthermore, a previously unknown contextual bias between "bee" and "flower" in ImageNet was discovered by B2T.

3. **Performance Improvement:** B2T outperforms other unsupervised models in discovering biased keywords. Furthermore, the debiased B2T-DRO classifier achieves a higher worst-group accuracy over other classifiers, such as a debiased DRO classifier using ground-truth bias labels.

4. **CLIP zero-shot prompting:** B2T can generate prompts using the detected bias keywords for better worst-group and average CLIP zero-shot classification performance, compared to the original CLIP templates combined with the bias group label. Additionally, non-bias keywords decrease it.

5. **Label Diagnosis:** The B2T keywords can be used to identify wrong and ambiguous labels.

## 3 Methodology

This section describes our approach to reproducing the work of Kim et al. (2024). An overview of the B2T framework is given in subsection 3.1. Subsections 3.2, 3.3, and 3.4 describe models, datasets, and hyperparameters respectively. The experimental setup and code, and the computation requirements are subsequently described in subsections 3.5 and 3.6. The author's codebase is publicly available, however, it is not complete enough to replicate all findings. We discuss which parts of the code were incomplete and our approach to implementing them based on the original paper. Afterward, the methodology for the additional experiments of our own is presented.

### 3.1 The B2T framework

The framework relies on a classifier trained on the dataset. It starts with an image captioning model ClipCap (Mokady et al., 2021). This model generates captions for a dataset of images. Common keywords are extracted from the generated captions of images misclassified by the classifier with the YAKE algorithm (Campos et al., 2020). Then the CLIP score is applied to measure the similarity of the keywords to the mispredicted images, to validate whether they represent biases or not (Radford et al., 2021). Subsection 3.2 outlines how this score is calculated.

After obtaining the bias keywords, they can be used for four different applications. First, to train a debiased version of the DRO classifier, named B2T-DRO. Second, for CLIP zero-shot classification prompting. The original prompt template is altered to include the keyword, this is further explained in subsection 3.2. The keywords can also be used to analyze and compare different classifiers, as well as to identify incorrect or ambiguous labels.

### 3.2 Model descriptions

**The CLIP score.** The CLIP score measures the similarity between a keyword of the generated caption to the actual contents of the image. A high CLIP score indicates a biased concept, whilst a low score suggests the opposite. It is calculated as depicted in equation 1 below:

$$s_{\texttt{CLIP}}(a; \mathcal{D}) = \texttt{sim}(a, \mathcal{D}_{wrong}) - \texttt{sim}(a, \mathcal{D}_{correct}). \tag{1}$$

$a$ is the keyword and $\mathcal{D}$ is the dataset, where $\mathcal{D}_{wrong}$ and $\mathcal{D}_{right}$ are the subsets of the validation set that correspond to the wrong and right predictions of the classifier respectively. Here, $\texttt{sim}(a, \mathcal{D})$ is the similarity computed as the average cosine similarity between the normalized CLIP embeddings of a keyword $f_{\texttt{text}}(a)$ and images $f_{\texttt{image}}(x)$ for $x \in \mathcal{D}$ according to equation 2:

$$\texttt{sim}(a, \mathcal{D}) := \frac{1}{|\mathcal{D}|} \sum_{x \in \mathcal{D}} f_{\texttt{image}}(x) f_{\texttt{text}}(a) \tag{2}$$

**B2T-DRO.** Kim et al. (2024) implement the *group DRO setting* with regularization proposed by Sagawa et al. (2020b). In this approach, the bias group with the highest loss within a training batch will determine the batch's overall loss, causing the DRO model to optimize its parameters specifically for this bias group. As per the original paper, a ResNet-50 model pre-trained on ImageNet is used as the starting point for training (He et al., 2016). The results in Sagawa et al. (2020a) suggest that the occurrence of differences in performance between groups in the data decreases by employing the DRO framework, negating the skew in biased datasets. B2T defines these groups by inferring sample-wise bias labels. This is achieved by adding the found bias keywords to the prompts of the CLIP zero-shot classifier, which is further explained below, and assigning the most probable B2T keyword to each image.

For evaluation, the keywords are then manually mapped to known bias labels to compare with the ground truth. The execution of this step is only possible for known biases and not previously unknown biases, such as any possible biases in ImageNet. Finally, every image should now have a bias label which allows the application of the group DRO framework.

**CLIP zero-shot classification.** The original authors aim to improve CLIP zero-shot classification by integrating biased keywords into the CLIP prompts. The original prompt template is: "a photo of a [class]." This is modified to: "a photo of a [class] in a [group]," where the keywords represent bias group names. Prompts are augmented with two distinct sets of keywords to assess their importance; B2T keywords with positive and negative CLIP scores. The authors have provided a template for prompt design with positive keywords, which can be found in appendix A. Unfortunately, the templates and keywords were not included for negative keywords.

The prompts are used to calculate new zero-shot weights for the pre-trained CLIP classifier and tested by predicting the most probable class. ResNet-50 was used as the image encoder. To measure the model's performance, the worst group accuracy and average accuracy are calculated for both prompt types. The

groups represent each class with its bias element, for example, landbird and water background. By comparing the results of the original prompts and keyword prompts, the effectiveness of the method can be evaluated.

### 3.3 Datasets

To identify known dataset biases and the applications of B2T keywords, we trained and tested the B2T framework on the CelebA and Waterbirds datasets. Our approach to analyzing biases without a classifier was also performed on these datasets. Both are included in the codebase with pre-trained model checkpoints.

**CelebA.** CelebA, which stands for CelebFaces Attributes, contains 202,599 images of celebrities' faces (Liu et al., 2015). Each image has 40 binary attribute annotations, such as blond and not blonde. The attribute "blonde" was employed as a binary target in Kim et al. (2024)'s work, with gender as the underlying bias label. This setting is biased against blonde males, who make up only 0.85% of the dataset compared to 14.05% for blonde females.

**Waterbirds.** Waterbirds is a dataset created by Sagawa et al. (2020b) consisting of artificially created images of cropped-out birds (Wah et al., 2011) transferred onto backgrounds of the "places" dataset by Zhou et al. (2018). The classes are the type of bird in the image: a waterbird or a landbird. Another attribute is the type of background of the image: a water background or a land background. Waterbirds has a total of 11.788 images. Within the dataset there is a bias against data points with conflicting backgrounds (data groups "waterbird on land" and "landbird on water"), as they collectively represent only 6% of the training set instead of the 50% one might expect in a balanced case.

**FairFace.** FairFace is a dataset created by Karkkainen & Joo (2021) consisting of 108.501 face images, collected from the YFCC-100M Flickr dataset. Afterward, the images were labeled with race, gender, and age groups. This dataset is used as an extension of the original project to investigate the generalization capabilities of the B2T bias extraction method. In this work, the "gender" label is used as a classification target, instead of an underlying bias, to ensure enough variation from the CelebA dataset. The "race" and "age" labels were independently chosen as underlying biases to reflect real-world use cases, such as in the work of Zhao et al. (2021). To replicate the design of the Waterbirds and CelebA dataset, two race values were selected to ensure a binary bias label. "White" and "Black" were chosen. Thus, the groups in the dataset are "black female", "black male", "white male" and "white female". Similarily, two age values were selected: "10-19" and "40-49". To further replicate the layout of the other datasets, these groups in the data are artificially downsampled. If a group is investigated in an experiment, the data concerning this group will be downsampled to a 3% ratio, similar to the ratios of the minorities in the waterbird dataset.

**ImageNet.** For identifying undiscovered dataset biases, the models were trained and tested on the ImageNet dataset. This is also included in our codebase. The used subset of ImageNet contains over a million images (Russakovsky et al., 2015). In the validation set, there are 50 images per class. This subset was used instead of the entire dataset due to limited computational resources.

### 3.4 Hyperparameters

The hyperparameter values specified by Kim et al. (2024) were used to reproduce their work. Batch size was changed however to 512, as we noticed the original authors employed a batch size of only one during captioning. In section 4 we further reveal the resulting difference in GPU usage and captioning time. The original paper does not include any seed values employed to run the models, so we enhance future reproducibility by including the seeds employed: 32, 16, and 8 for DRO and DRO-B2T on CelebA and Waterbirds. Our codebase includes the test performance of all seeds.

### 3.5 Experimental setup and code

The authors have made part of the code publicly available.[1] However, this is missing the implementation of several experiments that support their key claims. In this subsection, we outline our process for re-implementing these experiments and conducting our own additional experiments. To ensure accuracy, the

---

[1]https://github.com/alinlab/b2t

paper of the authors was followed as closely as possible during the re-implementation process. Our code is also publicly available.[2]

**CLIP score validation.** To test the validity of the CLIP score, the first claim mentioned in section 2, an ROC graph will be deployed for a set of neutral and biased keywords to obtain the AUROC score. Specifically, five keywords from the waterbirds dataset. This figure depends on the availability of the CLIP similarity per image for every chosen keyword, and their corresponding CLIP scores. The ROC is established by evaluating whether the similarity value of a keyword to an image can determine if the image is unbiased. The AUROC score indicates keyword bias, with lower scores suggesting biases. This is because biased keywords indicate the presence of biased images, whilst the ROC curve is focused on determining unbiased ones.

**Bias identification.** The authors have provided code for identifying bias keywords with the B2T framework for the Waterbirds and CelebA datasets, including pre-trained model checkpoints. This is however not the case for ImageNet and Dollar Street. The original paper described discovering unknown biases in these datasets, thus we re-implemented this method for ImageNet. We aimed to minimize resource usage, so Dollar Street was excluded as verifying the main claims does not depend on it. For these datasets, we use the ResNet50 with model weights IMAGENET 1K Version 1 from the standard PyTorch library, as mentioned in Kim et al. (2024).

**B2T-DRO.** The code for training the DRO and B2T-DRO debiased classifiers was provided by the authors. This is performed on the CelebA and Waterbirds datasets. To assess the third claim, we compared the worst-group and average accuracy of the B2T-DRO to six other classifiers that were referenced and utilized from previous studies in the original paper; ERM Sagawa et al. (2020b), LfFNam et al. (2020), GEORGESohoni et al. (2022), JTTLiu et al. (2021), CNCZhang et al. (2024) and aforementioned DRO. Afterward, the authors extracted bias keywords of ERM and DRO with CLIP scores higher than one, and their gap. They check whether these keywords are still present after debiasing. If they are, the CLIP score is now lower.

**CLIP zero-shot prompting.** Regarding the fourth claim, the original paper does describe their methodology for CLIP zero-shot prompting. However, in order to re-implement it, partial code from other files had to be extracted and fit to the experiment. The creation of zero-shot weights with the text prompts is based on the code for group label interference and the calculation of worst-group accuracy from the group DRO testing. As mentioned in subsection 3.2, the prompt template for positive bias keywords was included in the original paper and can be directly copied for reproduction. This is not the case for negative keywords, hence we constructed our own based on the format of the positive templates and bias keywords with negative CLIP scores as reported in the appendix of Kim et al. (2024). The experiment is done on the test set of the Waterbirds dataset. The CelebA dataset has been left out due to time constraints.

**Label diagnosis.** The methodology for performing label diagnosis is not described in the original paper. Thus, to test the validity of this claim, we examined a selection of bias keywords provided by the authors: Boar, Bee, Desk and Market. The Desk and Market category have significantly more potentially mislabeled images, making it inefficient to inspect manually. Besides the size, these categories are not suitable for manual search as they are often ambiguous; meaning that images where the keyword and class do not match may still not truly be misclassified. For example, an image with bell peppers in a market stand contains both bell peppers and a market, hence neither label could be considered incorrect. Thus, we only manually searched through the boar and bee category. Additionally, these classes were selected because the pig and warthog classes, as well as the ant class, overlap significantly with the boar and bee classes, respectively.

First, all image captions and predictions that contain one of the selected keywords but belong to a different class are selected. For instance, an image with a caption containing "bee" that is predicted as "bee," yet is labeled differently, would be selected. Then, we manually investigate the selected pictures for the stated classes to compare the effectiveness of this method to a fully manual analysis.

**Code optimization.** Our first extension was to optimize the image captioning code by increasing the batch size from one to 512, and removing unnecessary computation inefficiencies. For more details about the implemented changes, see appendix F. A comparison is made between the code provided by the authors and our implementation. To measure the computational speed, we track how long it takes to generate a

---

[2]https://github.com/Joeyjdl/b2t-reproduction

single caption, and the total captioning time for the entire dataset. Based on this, the improvement factor is calculated by dividing the per caption time of the original code by the optimized version. To evaluate the GPU usage, we simply check how much GPU memory is utilized.

**New dataset generalizability.** For our second extension, we apply the pipeline of B2T bias extraction on the FairFace dataset described in 3.3 to test its generalizability. This builds further upon the limitations discussed by Kim et al. (2024); the captioning and scoring models are potentially biased, as ClipCap and CLIP as less effective for specialized domains (Mo et al., 2023). Kim et al. (2024) applied B2T to the ChestX-ray14 Wang et al. (2017) and FMoW (Christie et al., 2017) datasets, consisting of medical and satellite images respectively. They concluded that ClipCap generates nonsensical captions for these datasets and that a specialized captioning model is needed to apply B2T effectively. Our experiment aims to determine how well B2T can be applied to a less specialized dataset like FairFace, which still differs from the original datasets as described in subsection 3.3.

An experiment is performed for every group in the data, where said group becomes a 3% minority in the training data. We then train a separate ERM classifier for each minority group. Training a regular ERM classifier on this data causes the accuracy for the minority group to be $\pm$ 30% lower than the average accuracy, allowing the extraction of biases using the mispredictions. We can confirm that the B2T pipeline generalizes to this dataset, if the extracted bias keywords overlap with the age or race of the investigated minority group. For example, "old" in the "40-49" age-related case or "black" when black people are the chosen minority. Unlike the original paper, we consider all groups as a potential minority to fully maximise the scope of our investigation.

## 3.6 Computational requirements

The B2T pipeline, without training a classifier, was run on a T4 GPU available via Google Colab. Captioning the CelebA validation set, extracting keywords, and calculating the CLIP scores takes around 30 minutes. For the smaller Waterbirds dataset, this process was performed in 10 minutes. The computationally heavy DRO training was run on an A100 GPU on Snellius, which consumes 512 SBUs per GPU hour for a full node (SURF User Knowledge Base, 2025). Training a DRO model on the CelebA dataset on the given 50 epochs took 4 hours, whilst training the same DRO model for 300 epochs on the Waterbirds dataset took 2 hours. The total SBU's spent to train all models for 3 seeds is about 5542 SBU's, equivalent to 10.8 GPU hours. Using the 2024 carbon intensity of the Netherlands, which is 0.37 kCO$_2$eq/kWh (Nowtricity, 2025), and the PUE of the SURF HPC datacenter equal to 1.2 (SURF, 2017), we calculate the total emissions according to equation 3:

$$CO_2e = CI * PUE * P * t \tag{3}$$

The A100 GPU has a maximum power consumption of 0.25kW, and the AMD EPYC 9934 CPU is at 0.21 kW (NVIDIA, 2020). Leading to a carbon footprint of 2.21 kCO$_2$e. The T4 has a power consumption of 0.70kW (NVIDIA, 2021). We have approximately spent 20 hours on this GPU, adding 0.6216 kCO$_2$e for a total of 2.83 kCO$_2$e.

# 4 Results

In this section, we outline the results obtained for each claim and the extent of which they correspond with the original paper. Also described are the results of our own experiments to extend the authors' framework.

## 4.1 Results reproducing original paper

### Result 1 - CLIP score validation

The first claim is that the CLIP score is a valid method to determine whether a keyword represents a bias or not. To corroborate this, the CLIP score of five different keywords are depicted in figure 1a, as well as the ROC graph of neutral and biased keywords in figure 1b. Finally, the correlation between the CLIP score and the AUROC is displayed in figure 1c. Our reproduced findings correspond with those of the original authors, depicted in the figure. 2. The reproduced results show that biased keywords in figure 1b have a

lower AUROC score. Furthermore, we observe that the CLIP scores in general negatively correlate with the AUROC score. Further reproduced results with similar sentiments are available in appendix D. These findings support the first claim.

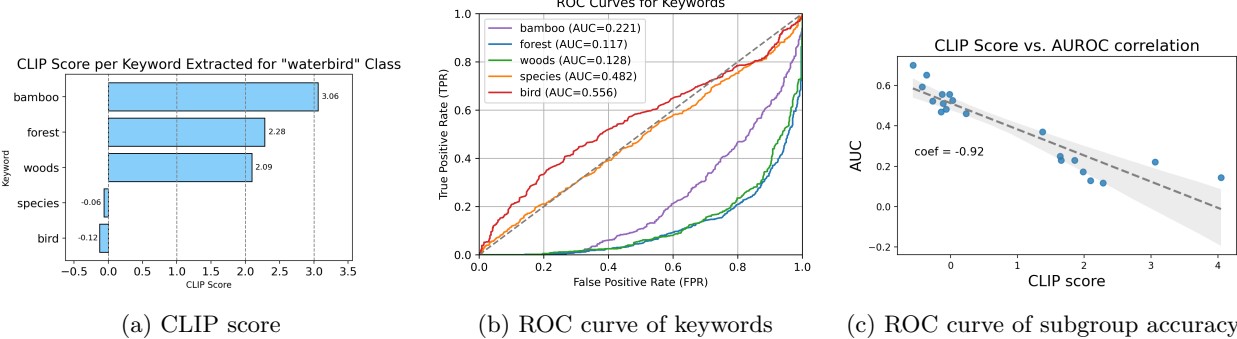

(a) CLIP score  (b) ROC curve of keywords  (c) ROC curve of subgroup accuracy

Figure 1: Effect of the CLIP score on non-bias and bias words for "waterbird" class (reproduction)

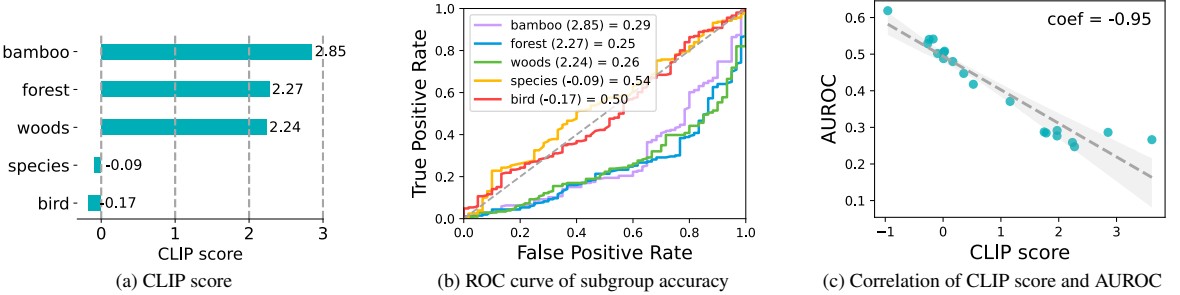

(a) CLIP score  (b) ROC curve of subgroup accuracy  (c) Correlation of CLIP score and AUROC

Figure 2: The CLIP score validation results of Kim et al. (2024) for "waterbird" class

**Result 2 - Bias identification**

The second claim of the original paper was that the extracted bias keywords relate to known ground truth bias labels and visual concepts that cause biases in ImageNet. Table 1 shows examples of keywords extracted using the B2T framework, combined with mispredicted images where the caption included the keyword.

The found keywords align with known biases in Waterbirds and CelebA. For instance, when an image from Waterbirds is wrongly predicted, the keyword "water" is found. Furthermore, in the ImageNet entries, we observe keywords that point to previously undiscovered biases. For example, "flower" appears as a bias keyword where an ant was misclassified as a bee. Other examples include more abstract keywords, such as the keyword for a mispredicted illustration of an ant being "white vector illustration". These likely stem from errors of the captioning model, resulting in unusual captions and, consequently, abstract keywords. In appendix B the original examples from the authors' paper are included for comparison. The original image paths were not given, so we chose fitting ones ourselves. To conclude, the keywords match the results of the original paper.

**Result 3 - Performance Improvement**

Two parts of the B2T pipeline are investigated for performance improvements. First, the bias discovery method is compared to three other models by Kim et al. (2024). Figure 3a shows their comparison for the CelebA dataset and figure 3c for Waterbirds by deploying a ROC graph. We successfully re-implemented these methods, as shown in figure 3b for CelebA and 3d for Waterbirds, and obtained similar results as Kim et al. (2024). We included the two best-performing baselines.

Furthermore, the original paper employed the discovered bias labels to debias the classifier used to obtain mispredictions. The original and reproduced worst-group and average accuracies after debiasing the classifier

| (a) Waterbirds | | | | (b) CelebA |
|---|---|---|---|---|
| **Keyword** | Water | Forest | Trees | Man |
| **Samples** |  |  |  |  |
| **Actual** | landbird | waterbird | waterbird | blond |
| **Pred.** | waterbird | landbird | landbird | not blond |
| **Caption** | a bird in the **water**. | the dinosaur was found in a **forest**. | a bird in the **trees** | actor - the face of a **man** who is always looking for a new adventure. |
| (c) ImageNet | | | | |
| **Keyword** | Glass jar | White vector illustration | Flower | Stormy sky |
| **Samples** |  |  |  |  |
| **Actual** | axolotl | ant | ant | armadillo |
| **Pred.** | newt | french horn | bee | lakeside, lakeshore |
| **Caption** | a spider in a **glass jar**. | black and **white vector illustration** of a black and white worm. | a bee on a **flower**. | a **stormy sky** over a river. |

Table 1: Examples from the Waterbirds, CelebA, and ImageNet datasets showing the ground truth and predicted labels, as well as captions for each keyword and sample.

are presented in table 3. We also included the best-performing baseline from the original paper for comparison. In appendix C, table 12, the performance with all original baselines is included. Our reproduced results show that B2T-DRO outperforms the strongest baseline in worst-group accuracy. These findings align with the authors' original results, further solidifying the third claim.

| | Keyword | ERM | B2T-DRO | Gap |
|---|---|---|---|---|
| **CelebA Blond** | man | 1.28 | × | × |
| **Waterbird** | bamboo forest | 4.04 | 3.06 | -0.98 |
| | bamboo | 3.06 | 2.43 | -0.63 |
| | forest | 2.28 | 1.75 | -0.53 |
| | woods | 2.09 | 1.64 | -0.45 |
| **Landbird** | seagull | 2.46 | 1.34 | -1.12 |
| | beach | 2.57 | 1.10 | -1.47 |
| | water | 1.26 | 1.04 | -0.22 |
| | lake | 1.04 | 1.15 | +0.11 |

Table 2: Reproduced model comparison: CLIP scores for keywords using ERM vs B2T-DRO. We mark × if the bias keyword is not found.

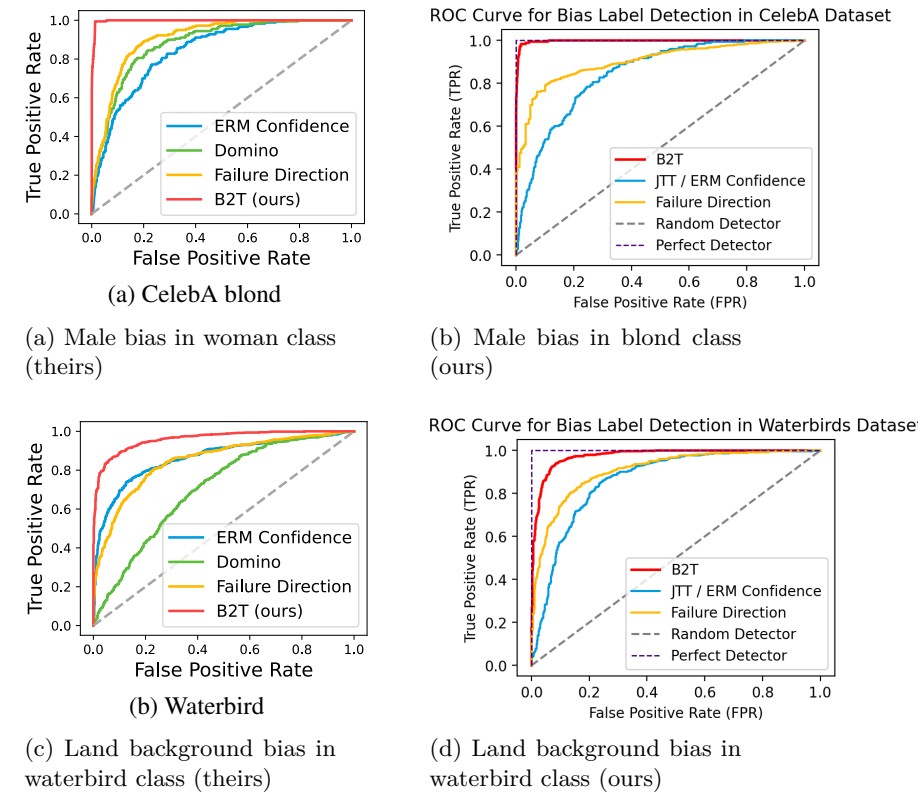

(a) Male bias in woman class (theirs)

(b) Male bias in blond class (ours)

(c) Land background bias in waterbird class (theirs)

(d) Land background bias in waterbird class (ours)

Figure 3: ROC graphs of bias detection methods on CelebA and Waterbirds datasets.

Lastly, table 2 depicts the CLIP scores for the bias keywords using ERM and B2T-DRO. Here, the CLIP scores are indeed lower after debiasing, further solidifying the third claim. However, all bias keywords except for 'man' still appear in the datasets, which deviates from the findings of the original authors. Their table is included in Appendix C, table 13.

| Method | CelebA blond | | Waterbirds | |
|---|---|---|---|---|
| | Worst | Avg. | Worst | Avg. |
| CNC [3](best baseline) | $88.8 \pm 0.9$ | $89.9$ | $88.5 \pm 0.3$ | $90.9$ |
| DRO with GT (ours) | $90.2 \pm 0.9$ | $93.1 \pm 0.2$ | $88.2 \pm 0.3$ | $92.5 \pm 0.3$ |
| DRO-B2T (ours) | $\mathbf{90.4} \pm 0.5$ | $93.0 \pm 0.1$ | $\mathbf{88.6} \pm 0.1$ | $90.7 \pm 1.0$ |
| DRO with GT (theirs) | $90.0 \pm 1.5$ | $93.3$ | $89.9 \pm 1.3$ | $91.5$ |
| DRO-B2T (theirs) | $\mathbf{90.4} \pm 0.9$ | $93.2$ | $\mathbf{90.7} \pm 0.3$ | $92.1$ |

Table 3: Comparison of DRO with GT and DRO-B2T (ours) vs. (theirs) and CNC baseline on the CelebA blond and Waterbirds datasets.

**Result 4 - CLIP zero-shot prompting**

The results for the CLIP zero-shot prompting experiment are displayed in table 4. The results utilizing the positive keywords are close to the original scores, but do not align exactly. Results applying negative keywords vary more, with our worst-group accuracy being around 10% higher.

---

[3]Zhang et al. (2024)

|  | Waterbirds | |
|---|---|---|
|  | Worst | Avg. |
| CLIP zero-shot | 50.3 | 72.7 |
| + Group prompt Wu et al. (2019) | 53.7 | 78.0 |
| + B2T-neg prompt(ours) | 54.3 | 74.4 |
| + B2T-pos prompt (ours) | **59.7** | 76.4 |
| + B2T-neg prompt (theirs) | 45.4 | 70.8 |
| + B2T-pos prompt (theirs) | 61.7 | 76.9 |

Table 4: worst-group and average accuracy comparison of different prompting methods for zero-shot classification

**Result 5 - Label Diagnosis**

According to Shankar et al. (2020), there are label errors in ImageNet. When manually looking at the class images of warthog, bee, pig, boar, and ant, there is an average of two mislabeled images per class. Generalizing this means that two in 50 images are mislabeled per class.

The results of the label diagnosis are shown in table 5. We observe that the label diagnosis method reduced the search to 39 for the boar class. Of these potentially mislabeled images, only two are actually mislabeled. One appeared in the selection by chance; it has the ground-truth label 'polar bear', but is mislabeled as a boar. By manually reviewing the similar 'hog' and 'pig' classes, we found three additional mislabeled boar images not included in the selection.

| Category | Number of Potential Images | ImageNet Class | Correct Mislabeled Images Found |
|---|---|---|---|
| Boar | 39 | Boar | 1 |
| Bee | 26 | Bee | 2 |
| Desk | 116 | Desktop Computer | Not Tested |
| Market | 215 | Grocery Store | Not Tested |

Table 5: Pottentially misclassified images, their descriptions, and actual mislabeled images found

## 4.2 Results beyond original paper

**Additional Result 1 - Optimization** Table 6 shows the computational speed of the original code and the optimized code. An improvement factor of around six times for the time taken for captioning is achieved.

| Version | Per Caption Time (s) | Total Time (s) | GPU Memory Utilized | Improvement Factor (Based on Caption Time) |
|---|---|---|---|---|
| Original | $0.1195 \pm 0.0057$ | $143.25 \pm 6.85$ | ~2GB of 15GB | 1 (Baseline) |
| Optimized | $0.0191 \pm 0.0007$ | $22.9 \pm 0.85$ | ~12GB of 15GB | $6.26 \pm 0.38$ |

Table 6: Performance comparison between original and optimized versions (run on Google Colab using T4 GPU)

| FairFace(race) Visualisation | | |
|---|---|---|
| **Samples** |  |  |
| **Labels** | man(black) | man(white) |
| **Caption** | a **black** man smiles at the camera. | a man with a beard and mustache. |

Table 7: Examples from the FairFaces dataset showing target and race labels, along with their generated captions. Bold letters indicate a bias keyword.

**Additional Result 2 - New dataset generalizability** Table 7 provides representative FairFace samples to clarify the upcoming results. The additional results concerning the generalization of B2T to racial bias and age bias are presented in table 8 and table 9 respectively. Furthermore, the top five extracted keywords with the highest CLIP score are shown for every minority, similar to how the original authors displayed CelebA and Waterbirds keywords. The final column indicates whether the keywords correspond to the underlying biases. For the "black female" group, the detected keyword "long black" can be associated with the underlying bias. However, no racial keywords were found for the other three race-based minorities.

For the age-related cases, we further observe some valid cases. Some detected keywords for the "10-19 age" cases, such as "young" and "young boy" in the case of males, are valid bias keywords. Keywords indicated with a "∼" are considered as rough correspondences to the bias in question, depending on the interpretation. For example, in the aforementioned male case, "boy" can be considered valid as it may correspond to a younger male, depending on the context in which the word is used. "young girl" is also considered a blurry case for the male minority, as "girl" is invalid but "young" is valid. Finally, we do not observe any valid bias keywords concerning the "40-49 age" cases.

To investigate the lack of bias keywords found for certain groups, we analyzed the frequency of available "ground truth" values in the captions, such as "old" for any of the "40-49" minorities. Results show that bias keywords pertaining to "black" and "young" are about three times more common in the captions than their respective counterparts, "white" and "old". These counterparts are the minorities for which there are no valid keywords found.

| Minority | Keyword | Bias Score | Bias |
|---|---|---|---|
| White male | girl | 1.69 | No |
| | child smiles | 1.44 | No |
| | child | 1.38 | No |
| | woman | 1.08 | No |
| | young boy smiles | 1 | No |
| White female | boy | 2.09 | No |
| | portrait | 0.38 | No |
| | found | 0.27 | No |
| | head | 0.20 | No |
| | person | 0.09 | No |
| Black male | child smiles | 1.77 | No |
| | girl smiles | 1.64 | No |
| | girl | 1.53 | No |
| | child | 1.50 | No |
| | young boy smiles | 1.42 | No |
| Black female | child | 2.42 | No |
| | boy | 2.11 | No |
| | long black | 1.50 | Yes |
| | glasses | 1.16 | No |
| | born | 1.06 | No |

Table 8: Racial bias keywords analysis

| Minority | Keyword | Bias Score | Bias |
|---|---|---|---|
| Male, age 10–19 | young girl | 3.17 | ∼ |
| | young boy | 2.75 | Yes |
| | young boy smiles | 2.31 | Yes |
| | girl | 2.22 | No |
| | boy | 2.19 | ∼ |
| Female, age 10–19 | boy | 1.78 | No |
| | man | 1.31 | No |
| | boy smiles | 0.94 | No |
| | young | 0.61 | Yes |
| | born | 0.50 | No |
| Male, age 40–49 | girl | 1.91 | No |
| | girl smiles | 1.81 | No |
| | woman | 1.48 | No |
| | face painted | 1.11 | No |
| | young boy smiles | 0.98 | No |
| Female, age 40–49 | man | 1.34 | No |
| | boy | 1.14 | No |
| | police | 0.63 | No |
| | found unconscious | 0.56 | No |
| | shocked | 0.45 | No |

Table 9: Age bias keywords analysis

# 5 Discussion

In this work, we successfully reproduced the first three main claims of the authors, and partially reproduced the fourth, but were unable to reproduce the final claim. We optimized the original code, and tested the generalizability of B2T on a new dataset with diverse biases.

The results for the fourth claim corroborate that the B2T-positive keywords improve the worst-group and average accuracy of zero-shot inferences. However, B2T-negative keywords do not reduce the accuracy as the original paper states. Slight differences are most likely due to missing implementation details, as explained in section 3. Similarly, we were not able to reproduce the fifth claim. We found that it fails to identify all mislabeled images and does not significantly reduce the search space of potential labeling errors. This could be caused by ambiguous or incorrect captions, making it difficult to reliably filter mislabellings based on the mismatch between image labels and caption keywords. Further improvements could explore how label diagnosis performance changes with enhanced captioning and more reliable keywords. It is also important to note that these results are based on limited manual searches; further investigation is needed to fully identify the limitations of this method.

For our extensions, captioning computes around six times faster after optimizing the code, which is a significant leap in compute time. When testing the generalizability of B2T on FairFace, the model was unable to consistently find a valid bias keyword for a majority of the age and race bias groups. This indicates an inability of the B2T framework to generalize to the characteristics of the FairFace dataset and possibly other datasets. This failure could be caused by the captioning model. As discussed in section 4, the minorities where valid keywords were found had a disproportionate amount available within the captioning data. This points towards biases of the captioning model in this race and age scenario, which could be alleviated by employing a more descriptive captioning model. Biases within the captioning model, such as only mentioning race when the individual is non-white, can be perpetuated in the keywords found by B2T. Since the examined biases are highly relevant, as systematic discrimination based on age or race is unwanted, further research is necessary to properly determine the generalizability limitations of B2T beyond the scope of a reproducibility paper.

In conclusion, the original paper aimed to introduce a novel pipeline to detect and mitigate visual biases through keyword explanations. Its methodology aligns with broader efforts to ensure fairness in datasets. However, we encountered challenges in reproducing the results, such as missing code and differing results, highlighting the need for accessible implementations. In the following subsections, we discuss which parts of the original work were easy to reproduce in 5.1 and which were hard in 5.2. Lastly, we discuss the extent of communication with the original authors in appendix H.

## 5.1 What was easy

The B2T framework is described well in the paper. It is clearly outlined which steps the framework consists of and explains what each step entails. Furthermore, the literature background in the original paper made it easy to find related papers to broaden our knowledge concerning the topic and inspire us to consider possible extensions. Additionally, a method of evaluating each claim in the paper is given. Because the code is publicly available, there is an initial framework for orientation and a basis for experiments.

## 5.2 What was difficult

An obstacle we faced during the reproduction of the paper was the incompleteness of the authors' code. For each of the main claims, some part of the code was missing. This includes 1) the preprocessing and implementation of the ImageNet dataset and the unused Dollar Street dataset, 2) the template for prompts with negative B2T keywords for CLIP zero-shot prompting, and 3) evaluation methods like the validation of the CLIP score. Smaller details, such as how often and on which seeds the debiased DRO classifiers were run to obtain the worst and average group scores, are also not included. Another issue was the lack of an ERM training file, as the codebase only included the pretrained ERM file. This meant reproducing the ERM results that were given in the original paper was not possible. Furthermore, due to the absence of an ERM file and time constraints, we had to implement the FairFace extension without training a classifier. As previously mentioned, this may degrade the academic value of this experiment.

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

# A    Prompt design templates

| Dataset-wise Template | Class Name |
|---|---|
| • [class name] on the forest
• [class name] with woods
• [class name] on a tree
• [class name] on a branch
• [class name] in the forest
• [class name] on the tree
• [class name] on the ocean
• [class name] on a beach
• [class name] on the lake
• [class name] with a surfer
• [class name] on the water
• [class name] on a boat
• [class name] on the dock
• [class name] on the rocks
• [class name] in the sunset
• [class name] with a kite
• [class name] on the sky
• [class name] is on flight
• [class name] is on flies | **1. Landbird**
• landbird

**2. Waterbird**
• waterbird |

Table 10: The template for debiased zero-shot prompting with positive keywords of Kim et al. (2024).

| Dataset-wise Template | Class Name |
|---|---|
| • [class name] eagle
• bald [class name]
• [class name] in the snow
• great [class name]
• large [class name]
• flying [class name]
• [class name] with a person
• [class name] in a pond
• biological [class name] species
• biological [class name]
• [class name] species in flight
• [class name] species
• [class name] bird | **1. Landbird**
• landbird

**2. Waterbird**
• waterbird |

Table 11: The template for debiased zero-shot prompting with negative keywords

# B    Original Keywords with Image Samples

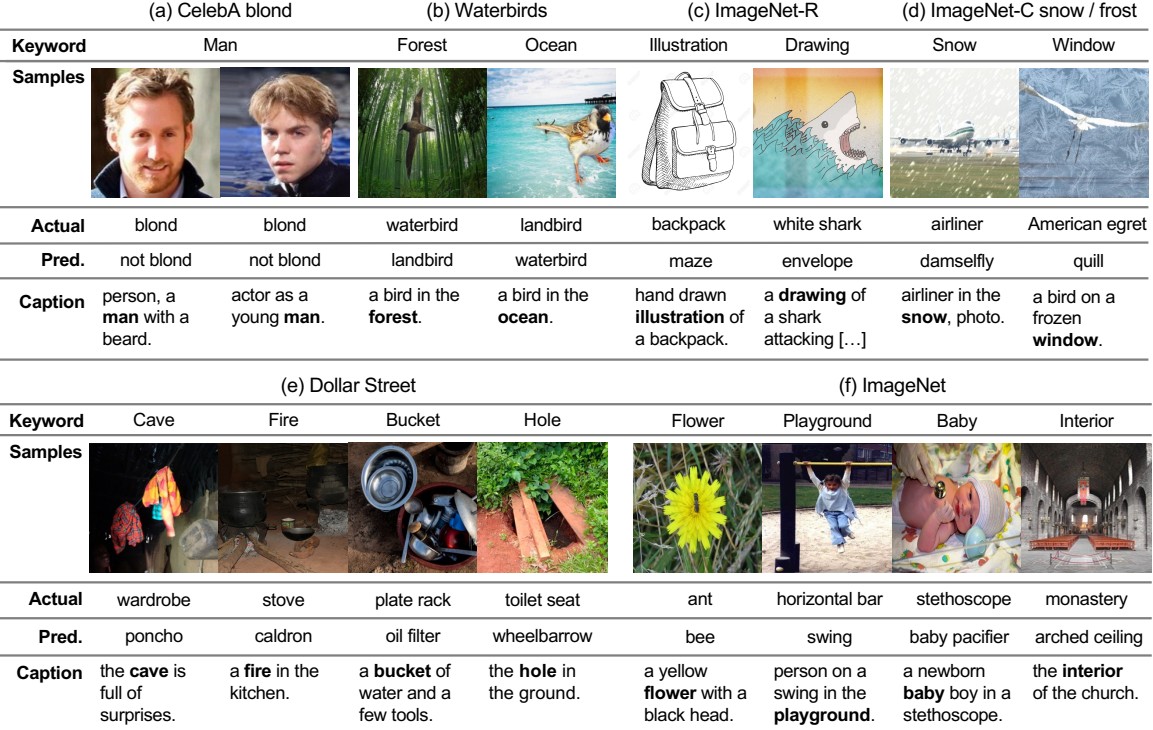

| | (a) CelebA blond | | (b) Waterbirds | | (c) ImageNet-R | | (d) ImageNet-C snow / frost | |
|---|---|---|---|---|---|---|---|---|
| **Keyword** | Man | | Forest | Ocean | Illustration | Drawing | Snow | Window |
| **Samples** | | | | | | | | |
| **Actual** | blond | blond | waterbird | landbird | backpack | white shark | airliner | American egret |
| **Pred.** | not blond | not blond | landbird | waterbird | maze | envelope | damselfly | quill |
| **Caption** | person, a **man** with a beard. | actor as a young **man**. | a bird in the **forest**. | a bird in the **ocean**. | hand drawn **illustration** of a backpack. | a **drawing** of a shark attacking [...] | airliner in the **snow**, photo. | a bird on a frozen **window**. |

| | (e) Dollar Street | | | | (f) ImageNet | | | |
|---|---|---|---|---|---|---|---|---|
| **Keyword** | Cave | Fire | Bucket | Hole | Flower | Playground | Baby | Interior |
| **Samples** | | | | | | | | |
| **Actual** | wardrobe | stove | plate rack | toilet seat | ant | horizontal bar | stethoscope | monastery |
| **Pred.** | poncho | caldron | oil filter | wheelbarrow | bee | swing | baby pacifier | arched ceiling |
| **Caption** | the **cave** is full of surprises. | a **fire** in the kitchen. | a **bucket** of water and a few tools. | the **hole** in the ground. | a yellow **flower** with a black head. | person on a swing in the **playground**. | a newborn **baby** boy in a stethoscope. | the **interior** of the church. |

Figure 4. **Discovered biases in image classifiers.** Visual examples of mispredicted images, along with their corresponding bias keywords, captions, actual classes, and predicted classes. B2T successfully identified known biases, such as (a) gender bias in CelebA blond, (b) background bias in Waterbirds, and distribution shifts in (c) ImageNet-R with different styles, and (d) ImageNet-C with natural corruptions. B2T also uncovered novel biases in larger datasets, such as the spurious correlations between (e) the keyword "cave" and the wardrobe class, indicating geographical bias in Dollar Street, and (f) the keyword "flower" and the ant class, indicating contextual bias in ImageNet.

Figure 4: Original paper's table of Kim et al. (2024) containing keywords accompanied with sample mispredicted images and their captions.

## C  Debiasing Results Tables

| Method | CelebA blond | | Waterbirds | |
|---|---|---|---|---|
| | Worst | Avg. | Worst | Avg. |
| ERM (excerpted from Zhang et al.) | $47.7 \pm 2.1$ | 94.9 | $62.6 \pm 0.3$ | 97.3 |
| LfF Nam et al. | 77.2 | 85.1 | 78.0 | 91.2 |
| GEORGE Sohoni et al. | $54.9 \pm 1.9$ | 94.6 | $76.2 \pm 2.0$ | 95.7 |
| JTT Liu et al. | $81.5 \pm 1.7$ | 88.1 | $83.8 \pm 1.2$ | 89.3 |
| CNC Zhang et al. | $88.8 \pm 0.9$ | 89.9 | $88.5 \pm 0.3$ | 90.9 |
| DRO with GT (ours) | $90.2 \pm 0.9$ | $93.1 \pm 0.2$ | $88.2 \pm 0.3$ | $92.5 \pm 0.3$ |
| DRO-B2T (ours) | $\mathbf{90.4} \pm 0.5$ | $93.0 \pm 0.1$ | $\mathbf{88.6} \pm 0.1$ | $90.7 \pm 1.0$ |
| DRO with GT (theirs) | $90.0 \pm 1.5$ | 93.3 | $89.9 \pm 1.3$ | 91.5 |
| DRO-B2T (theirs) | $\mathbf{90.4} \pm 0.9$ | 93.2 | $\mathbf{90.7} \pm 0.3$ | 92.1 |

Table 12: Comparison of various model debiasing methods, with all original baseline values and our reproduction

| | Keyword | ERM | B2T-DRO | Gap |
|---|---|---|---|---|
| **CelebA Blond** | man | 1.06 | $\times$ | $\times$ |
| **Waterbird** | bamboo forest | 3.61 | $\times$ | $\times$ |
| | bamboo | 2.85 | $\times$ | $\times$ |
| | forest | 2.27 | 1.97 | -0.30 |
| | woods | 2.24 | 1.88 | -0.36 |
| **Landbird** | seagull | 3.10 | 1.85 | -1.24 |
| | beach | 2.45 | 1.15 | -1.30 |
| | water | 1.51 | 0.67 | -0.84 |
| | lake | 1.25 | $\times$ | $\times$ |

Table 13: Original model comparison results of Kim et al. (2024): CLIP scores for keywords using ERM vs B2T-DRO. We mark $\times$ if the bias keyword is not found.

# D    CLIP Score Validation Results

This section includes further reproduction results of the CLIP score validation. These results (the correlations, the ROC curve) correspond to the ones found in section 4 for the waterbirds class, supporting the paper of the authors.

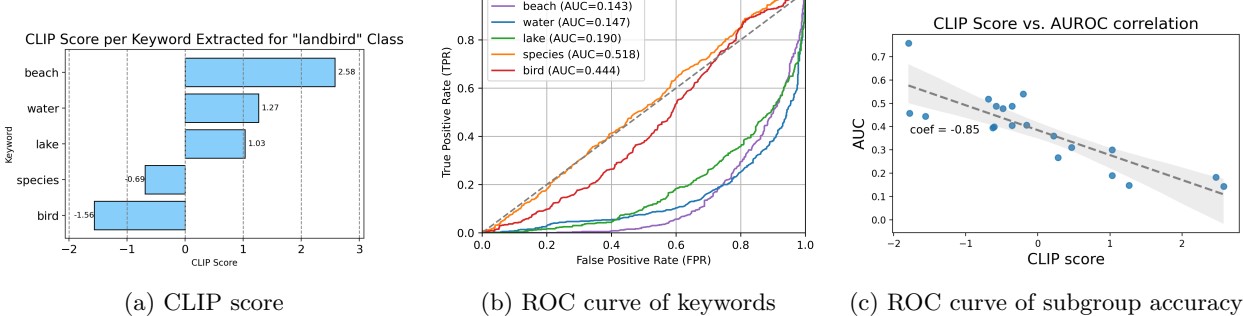

(a) CLIP score          (b) ROC curve of keywords          (c) ROC curve of subgroup accuracy

Figure 5: Effect of the CLIP score on non-bias and bias words for "landbird" class (reproduction)

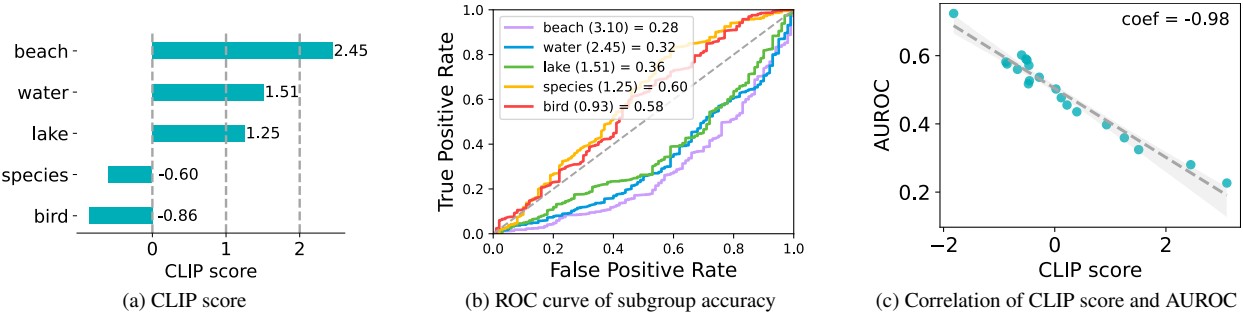

(a) CLIP score          (b) ROC curve of subgroup accuracy          (c) Correlation of CLIP score and AUROC

Figure 6: The CLIP score validation results of Kim et al. (2024) for "landbird" class

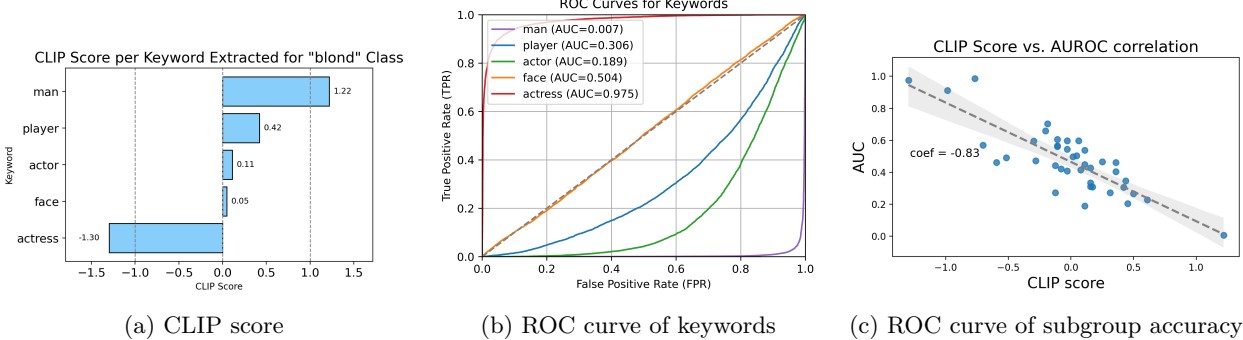

(a) CLIP score          (b) ROC curve of keywords          (c) ROC curve of subgroup accuracy

Figure 7: Effect of the CLIP score on non-bias and bias words for "blond" class in CelebA (reproduction)

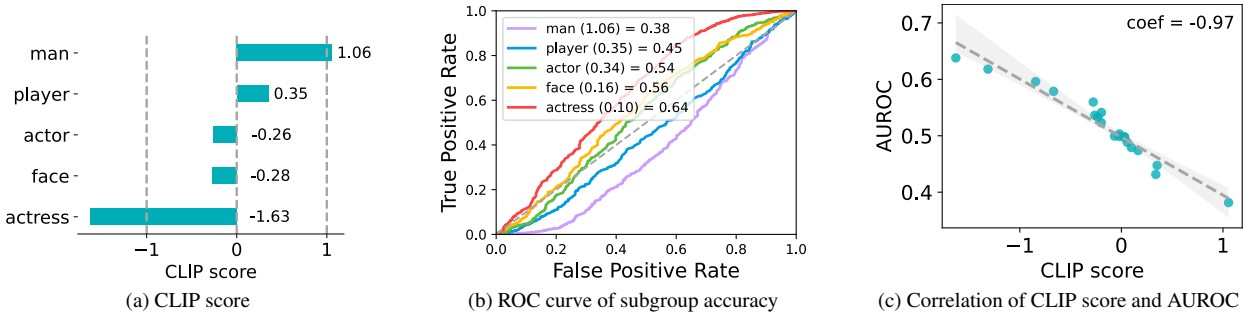

(a) CLIP score          (b) ROC curve of subgroup accuracy          (c) Correlation of CLIP score and AUROC

Figure 8: The CLIP score validation results of Kim et al. (2024) for "blond" class in CelebA

# E  Future direction: Bias Analysis without a Classifier

Future work could investigate the possibility of performing a bias analysis without a classifier. This could reduce the computational cost and environmental impact of the B2T pipeline, whilst still achieving the main goal of obtaining biases. As a proof of concept, we proposed a simple method where the B2T pipeline is followed, but the CLIP score is replaced with our SiMean score. The SiMean score does not utilize mispredictions as the CLIP score does. For a keyword $a$ in a set of extracted keywords $A$, the SiMean score is calculated according to equation 4:

$$\text{SiMean}(a, D) = \text{sim}(a, D) - \frac{1}{|D|} \sum_{k \in A} \text{sim}(k, D). \tag{4}$$

We applied this basic scoring system to the CelebA, Waterbirds and Fairfaces datasets. All extracted keywords and experimental details are available in the codebase notebook. The keywords found by this experiment do not exhibit specific bias characteristics like those found in the B2T keywords, though selected keywords do guide us to biases. For example: "flower" is extracted in the ImageNet "bee" class, and "background" is extracted for the "ant" class. These exact findings are also present in the examples from the B2T-guided keyword extraction method presented in table 1.

However, the general findings suggest that the scoring method is ineffective, as the highest-scoring keywords are often broad and uninformative. They describe the dataset by keywords that are most similar, instead of finding niche biases. Nonetheless, this approach demonstrates that eliminating the need for a computationally expensive classifier could make for a more time-efficient method that is less harmful to the environment. Further research beyond this reproduction, with a more advanced scoring and evaluation system, could exploit this. For example, future work could consider a method that compares sets of images without requiring an image classifier, possibly inspired from Dunlap et al. (2024) or Riccio et al. (2025).

# F  Technical details of code performance improvements

**Original Implementation**

- **extract_caption** processes each image sequentially.

- Per image there are three steps: **io.imread()**, conversion to PIL image, and CLIP preprocessing.

**Improved Implementation**

- **extract_caption_batch** makes use of **ThreadPoolExecutor** to parallelize the image preprocessing.

- I/O operations now happen concurrently, reducing wait time for GPU processing.

## F.1  Batched CLIP Embedding

**Original Implementation**

- Processes one image at a time through CLIP:

```
prefix = clip_model.encode_image(image).to(clip_device, dtype=torch.float32)
prefix_embed = caption_model.clip_project(prefix).reshape(1, prefix_length, -1)
```

**Improved Implementation**

- Preprocessed images are stacked into a single tensor batch and than run through CLIP as a batch:

```
batch_prefix = clip_model.encode_image(batch_tensor).to(clip_device, dtype=torch.float32)
batch_prefix_embed = caption_model.clip_project(batch_prefix).reshape(len(batch_paths), prefix_length,
    -1)
```

- This efficiently utilizes the GPU through batch processing

### F.2 Batched Caption Generation

**Original Implementation**

- **generate2** generates a single token at a time.

**Improved Implementation**

- **generate2_batch** generates a batch of tokens at once.

- Tracks completed and not completed sequences with boolean mask:

```
is_stopped[original_indices] |= (next_token == stop_token_index)
active_indices = ~is_stopped[original_indices]
```

- Remove complete captions from the generation batch:

```
generated = generated[active_indices]
```

- Removed unnecessary data copying to reduce memory usage by a factor of 32.

- Efficient compute usage by only generating tokens for unfinished captions.

- Use tokenizer's **batch_decode** for decoding captions in a batch.

```
decoded_captions = tokenizer.batch_decode(tokens.cpu().numpy(), skip_special_tokens=True)
```

## G   Extended FairFace Generalizability Results

| Male, Age 10–19 | | | Female, Age 10–19 | | |
|---|---|---|---|---|---|
| Keyword | Bias Score | Bias | Keyword | Bias Score | Bias |
| young girl | 3.17 | ∼ | boy | 1.78 | No |
| young boy | 2.75 | Yes | man | 1.31 | No |
| young boy smiles | 2.31 | Yes | boy smiles | 0.94 | No |
| girl | 2.22 | No | young | 0.61 | Yes |
| boy | 2.19 | ∼ | born | 0.5 | No |
| boy smiles | 1.56 | ∼ | head | 0.47 | No |
| young | 1.30 | ∼ | student | 0.39 | No |
| student | 1.25 | No | nose | 0.39 | No |
| boys | 1.20 | ∼ | camera | 0.36 | No |
| genetic condition | 1.13 | No | young girl | 0.28 | Yes |
| rare genetic condition | 1.06 | No | village | 0.28 | No |
| born | 0.98 | No | long | 0.27 | No |
| rare genetic | 0.94 | No | tooth | 0.23 | No |
| missing tooth | 0.88 | No | hair | 0.20 | No |
| tooth | 0.88 | No | condition | 0.19 | No |
| woman | 0.86 | No | person | 0.19 | No |
| genetic | 0.78 | No | shot | 0.17 | No |
| young man | 0.61 | Yes | face | 0.16 | No |
| smile | 0.58 | No | found | 0.14 | No |
| smiles | 0.52 | No | rare | 0.14 | No |
| shot | 0.47 | No | broken | 0.14 | No |
| rare | 0.38 | No | broken nose | 0.11 | No |
| found | 0.33 | No | genetic condition | 0.09 | No |
| car | 0.30 | No | rare genetic | 0.06 | No |
| diagnosed | 0.25 | No | diagnosed | 0.05 | No |
| missing | 0.25 | No | missing | 0.03 | No |
| broken | 0.19 | No | missing tooth | 0 | No |
| face | 0.19 | No | genetic | 0 | No |
| punched | 0.13 | No | girl | -0.02 | No |
| condition | 0.08 | No | rare genetic condition | -0.02 | No |
| hair | 0.06 | No | bedroom | -0.03 | No |
| camera | 0.05 | No | portrait | -0.05 | No |
| bathroom | 0 | No | covered | -0.05 | No |
| head | -0.09 | No | bathroom | -0.06 | No |
| person | -0.11 | No | cancer | -0.08 | No |
| bedroom | -0.28 | No | young woman | -0.23 | No |
| arrested | -0.56 | No | woman | -0.31 | No |
| beard | -0.72 | No | young girl smiles | -0.34 | Yes |
| allegedly | -0.83 | No | smiles | -0.47 | No |
| man | -1.19 | No | girl smiles | -0.62 | ∼ |

Table 14: All extracted age bias keywords and corresponding bias scores for two minority groups: Male and Female, Age 10–19

| Male, Age 40–49 | | | Female, Age 40–49 | | |
|---|---|---|---|---|---|
| **Keyword** | **Bias Score** | **Bias** | **Keyword** | **Bias Score** | **Bias** |
| girl | 1.91 | No | man | 1.34 | No |
| girl smiles | 1.81 | No | boy | 1.14 | No |
| woman | 1.48 | No | police | 0.63 | No |
| face painted | 1.11 | No | found unconscious | 0.56 | No |
| young boy smiles | 0.98 | No | shocked | 0.45 | No |
| boy | 0.80 | No | nose | 0.45 | No |
| boy smiles | 0.75 | No | head | 0.45 | No |
| tooth | 0.72 | No | born | 0.42 | No |
| rare genetic condition | 0.70 | No | face was covered | 0.38 | No |
| smile | 0.70 | No | shot | 0.38 | No |
| genetic condition | 0.64 | No | glasses | 0.30 | No |
| smiles | 0.58 | No | rare | 0.28 | No |
| rare genetic | 0.53 | No | condition | 0.27 | No |
| born | 0.52 | No | tooth | 0.25 | No |
| painted | 0.5 | No | camera | 0.25 | No |
| shot | 0.5 | No | face full | 0.23 | No |
| genetic | 0.44 | No | person | 0.23 | No |
| face | 0.42 | No | rare genetic | 0.23 | No |
| rare | 0.42 | No | face | 0.22 | No |
| student | 0.42 | No | broken | 0.19 | No |
| native | 0.41 | No | named person | 0.17 | No |
| found | 0.34 | No | bedroom | 0.16 | No |
| young | 0.33 | No | found | 0.16 | No |
| missing | 0.25 | No | rare genetic condition | 0.16 | No |
| car | 0.23 | No | missing | 0.11 | No |
| house | 0.19 | No | genetic condition | 0.06 | No |
| person | 0.19 | No | diagnosed | 0.03 | No |
| hair | 0.19 | No | young | 0.03 | No |
| school | 0.17 | No | genetic | 0.02 | No |
| condition | 0.14 | No | missing tooth | 0.02 | No |
| bedroom | 0.13 | No | covered | -0.03 | No |
| camera | 0.09 | No | bathroom | -0.03 | No |
| head | -0.02 | No | student | -0.03 | No |
| found guilty | -0.23 | No | woman | -0.05 | ~ |
| guilty | -0.28 | No | portrait | -0.11 | No |
| arrested | -0.28 | No | smiles | -0.50 | No |
| young man | -0.30 | No | girl | -0.58 | No |
| beard | -0.38 | No | young girl | -0.69 | No |
| man smiles | -0.59 | ~ | smile | -0.69 | No |
| man | -0.80 | ~ | girl smiles | -0.89 | No |

Table 15: All extracted age bias keywords and corresponding bias scores for two minority groups: Male and Female, Age 40–49

| White male | | | Black male | | |
|---|---|---|---|---|---|
| **Keyword** | **Bias Score** | **Bias** | **Keyword** | **Bias Score** | **Bias** |
| girl | 1.69 | No | child smiles | 1.76 | No |
| child smiles | 1.44 | No | girl smiles | 1.64 | No |
| child | 1.38 | No | girl | 1.53 | No |
| woman | 1.08 | No | child | 1.50 | No |
| young boy smiles | 1.00 | No | young boy smiles | 1.42 | No |
| young boy | 0.92 | No | young boy | 1.15 | No |
| boy | 0.83 | No | boy smiles | 1.00 | No |
| boy smiles | 0.78 | No | boy | 0.89 | No |
| genetic condition | 0.69 | No | woman | 0.76 | No |
| smiles | 0.67 | No | tooth | 0.62 | No |
| rare genetic condition | 0.67 | No | smiles | 0.60 | No |
| pool | 0.58 | No | genetic condition | 0.57 | No |
| rare genetic | 0.58 | No | baby boy | 0.51 | No |
| tooth | 0.56 | No | rare genetic condition | 0.48 | No |
| shot | 0.52 | No | person smiles | 0.42 | No |
| water | 0.50 | No | born | 0.39 | No |
| bullet | 0.48 | No | young | 0.37 | No |
| genetic | 0.48 | No | bullet | 0.35 | No |
| born | 0.47 | No | student | 0.34 | No |
| student | 0.44 | No | boys | 0.34 | No |
| diagnosed | 0.41 | No | genetic | 0.34 | No |
| found | 0.41 | No | rare genetic | 0.29 | No |
| broken | 0.39 | No | diagnosed | 0.20 | No |
| missing | 0.38 | No | found | 0.14 | No |
| rare | 0.38 | No | condition | 0.12 | No |
| young | 0.34 | No | village | 0.09 | No |
| face | 0.28 | No | face | 0.07 | No |
| condition | 0.27 | No | home | 0.04 | No |
| home | 0.23 | No | shot | 0.03 | No |
| car | 0.23 | No | rare | 0.03 | No |
| person was found | 0.22 | No | car | 0.00 | No |
| camera | 0.19 | No | broken | -0.09 | No |
| bedroom | 0.17 | No | camera | -0.09 | No |
| hospital | 0.17 | No | person | -0.09 | No |
| head | 0.06 | No | fight | -0.10 | No |
| person | 0.05 | No | head | -0.15 | No |
| undated | -0.09 | No | bedroom | -0.21 | No |
| arrested | -0.16 | No | arrested | -0.39 | No |
| beard | -0.30 | No | beard | -0.50 | No |
| man | -0.70 | No | man | -0.82 | No |

Table 16: All extracted racial bias keywords and corresponding bias scores for two minority groups: White male and Black male

| Black female | | | White female | | |
|---|---|---|---|---|---|
| **Keyword** | **Bias Score** | **Bias** | **Keyword** | **Bias Score** | **Bias** |
| child | 2.42 | No | boy | 2.09 | No |
| boy | 2.11 | No | portrait | 0.38 | No |
| long black | 1.5 | Yes | found | 0.27 | No |
| glasses | 1.16 | No | head | 0.20 | No |
| born | 1.06 | No | person | 0.09 | No |
| long nose | 1.05 | No | head and neck | 0.03 | No |
| girl | 1.03 | No | bedroom | -0.22 | No |
| wearing glasses | 1.0 | No | shot | -0.28 | No |
| long | 0.95 | No | neck | -1.22 | No |
| black | 0.94 | Yes | | | |
| head and neck | 0.88 | No | | | |
| head | 0.84 | No | | | |
| neck | 0.83 | No | | | |
| hair wearing glasses | 0.83 | No | | | |
| condition | 0.81 | No | | | |
| nose | 0.81 | No | | | |
| camera | 0.78 | No | | | |
| diagnosed | 0.67 | No | | | |
| long black hair | 0.67 | ∼ | | | |
| mother | 0.64 | No | | | |
| found | 0.63 | No | | | |
| genetic condition | 0.59 | No | | | |
| black hair | 0.58 | ∼ | | | |
| resident | 0.56 | No | | | |
| genetic | 0.48 | No | | | |
| person | 0.47 | No | | | |
| rare genetic | 0.38 | No | | | |
| rare genetic condition | 0.36 | No | | | |
| person was found | 0.34 | No | | | |
| rare | 0.33 | No | | | |
| living | 0.31 | No | | | |
| bedroom | 0.30 | No | | | |
| hair | 0.30 | No | | | |
| shot | 0.25 | No | | | |
| bathroom | 0.17 | No | | | |
| portrait | 0.16 | No | | | |
| hair wearing | 0.03 | No | | | |
| wearing | -0.03 | No | | | |
| woman with long | -0.09 | No | | | |
| woman | -0.17 | No | | | |

Table 17: All extracted racial bias keywords and corresponding bias scores for two minority groups: Black female and White female

## H   Communication with original authors

An email was sent to the authors of the original paper to seek clarification on aspects of their methodology and results, pertaining to the missing information as discussed throughout this study. The authors informed us that they have deadlines that have not allowed them the time to answer our questions.

