# OpenReview forum: "Revisiting B2T: Discovering and Mitigating Visual Biases through Keyword Explanations"
_TMLR — Accepted by TMLR_

### Review · Reviewer_JPR8 · 2025-04-21

**Summary Of Contributions:**

This paper evaluates the claims of the original B2T (Bias-to-Text) work by Kim et al. (2024), which introduces a method for detecting and mitigating visual biases using keyword explanations extracted from image captions. The authors systematically assess five core claims from the original paper through both the provided codebase and their own re-implementations—confirming three claims, partially validating one, and refuting another. In addition, they contribute several extensions: an optimized captioning pipeline achieving a 6× speedup, an evaluation of B2T’s generalizability on a new dataset (FairFace), and the introduction of a novel bias scoring method, SiMean(a, D), which enables bias analysis without requiring a trained classifier.

**Audience:**

Yes

**Claims And Evidence:**

No

**Requested Changes:**

(1) Modification of SiMean Score

The idea of designing a bias scoring method that does not require an additional classifier is promising; however, the current SiMean score lacks the capacity to effectively discover biases. An alternative score design may be necessary. For instance, Dunlap et al., 2024 [1], proposes a method for comparing two sets of images without relying on a classifier, in a way that aligns with the goals of B2T. If a new score is introduced, a systematic evaluation will be essential to justify its effectiveness—an evaluation that is currently missing for the SiMean score. A basic quantitative comparison—such as overlap with ground-truth bias labels or performance impact when used in prompting—would help demonstrate its efficacy.


(2) Broader Expandability

To be a valuable contribution, the paper should go beyond faithfully reproducing the original framework and begin addressing some of its fundamental limitations. One major concern in the original B2T design is the possibility that both the captioning model (e.g., ClipCap) and the scoring model (e.g., CLIP) may inherit societal or dataset-induced biases. Another important limitation is that the B2T framework tends to fail in specialized domains (e.g., medical or satellite imagery) where models like CLIP were not originally trained. Tackling these issues and further investigating how B2T can be adapted or extended in such contexts would represent a promising and meaningful direction for future work.

[1] Dunlap et al., "VisDiff: Describing Differences in Image Sets with Natural Language, " 2024

**Strengths And Weaknesses:**

The authors provide a comprehensive reproduction of the original framework, supported by a fully re-implemented codebase. However, their additional contributions—(1) an optimized captioning pipeline, (2) an evaluation of generalizability on a new dataset, and (3) a novel bias scoring method that does not require a classifier—are somewhat limited in impact. Specifically, the captioning pipeline optimization amounts primarily to increasing the batch size from 1 to 512, which, while effective, is a relatively straightforward improvement. Moreover, in the FairFace generalization experiment, the use of manually simulated mispredictions rather than outputs from a trained classifier raises concerns about the validity and reliability of the reported findings. Lastly, the proposed SiMean(a, D) does not serve as an effective alternative to the original CLIP score used in B2T. While SiMean highlights “distinctive” keywords, these are not necessarily indicative of bias. This limitation arises because it overlooks the core idea of the original CLIP score that it *compares* the score between the misclassified and correctly classified images. As a result, SiMean does not catch actual model failure modes. Empirically, the keywords extracted using SiMean were often generic and uninformative (e.g., “person”, “face”), and frequently failed to reveal known biases, as the authors themselves acknowledge.

---

> ### Author Response · Authors · 2025-05-14
> **Official Response by Authors**
>
> Thank you for your insightful review and detailed feedback! We have addressed all your comments below and incorporated the relevant requested changes in our revision. The changed texts are currently blue for clarity within the revision.
>
> "The captioning pipeline optimization amounts primarily to increasing the batch size from 1 to 512, which, while effective, is a relatively straightforward improvement." We agree that this is a straightforward improvement, but we do find it of importance to mention due to the large improvement in computation. We have also added an appendix F with all additional code improvements outlined, which hopefully shows that the improvement achieved was realised by a combination of changes.
>
> "in the FairFace generalization experiment, the use of manually simulated mispredictions rather than outputs from a trained classifier raises concerns about the validity and reliability of the reported findings"
> We wholeheartedly agree that this raises clear concerns about the validity of the findings, and have thus trained a classifier separately for all minorities within the dataset, and updated all relevant parts accordingly. To further broaden the scope of our reproduction study, we have also extended the results for a different bias and added an appendix with additional results in response to the feedback of another reviewer.
>
> "Modification of SiMean Score" - We agree that the current SiMean score lacks the capacity to effectively discover biases. After careful consideration, we have moved any mention of this score to appendix E to ensure the validity of our reproduction is not compromised. In the appendix we also emphasize the limitations of this score, whilst highlighting the fact that it does show the B2T-pipeline could theoretically work without a classifier. Thank you for the recommendation of VisDiff, as it seems like a promising approach to this problem. The appendix incorporates a part based on the line of thought behind SiMean for future work outside the scope of reproduction, which includes a reference to VisDiff and another recent work.
>
> "Broader Expandability" - We agree that our contributions only partially consider the inherent limitations of B2T. Our focus within the reproduction is on validating the claims of the original authors, which we performed by employing B2T on the original datasets, and a dataset within a similar domain. This dataset should not be heavily affected by the limitations mentioned in the original paper, unlike other domains such as satellite imagery. This approach allows for a thorough validation of the original authors’ claims, without devaluing their work by shifting focus to unrelated domains. We used this approach, as this paper is meant as a reproducibility study for the Machine Learning Reproducibility Challenge (MLRC). The emphasis was therefore not on major contributions of our own, and we find the current contributions of code improvements and dataset generalizability sufficient to this end.
>
> We also made some changes according to the other reviewers. If you would like to see the specifics of this, please refer to the "Changes Since Last Submission" section. Finally, we would again like to thank you for your comprehensive review and constructive critique. Hopefully, we have successfully addressed your comments within our revision.

---

### Review · Reviewer_5VEM · 2025-04-28

**Summary Of Contributions:**

This paper reproduces and evaluates the results of a previous work, B2T. It validates the following claims: (i) the model can identify whether a word represents a bias; (ii) it can extract such keywords from captions of misclassified images; (iii) it outperforms other bias discovery models; (iv) it can improve CLIP zero-shot prompting using the discovered keywords; and (v) it can identify labeling errors in a dataset. The paper confirms the first three claims, partially supports the fourth, and rejects the fifth. In particular, it finds that negative prompts do not degrade CLIP’s zero-shot classification, and quantitatively shows that B2T does not effectively aid in label error diagnosis.

Additionally, the paper proposes several improvements: (i) bias analysis without a classifier using the SiMean score; (ii) codebase optimization to reduce CO₂ emissions; and (iii) further validation of the framework on a new dataset, FairFace.

**Audience:**

Yes

**Claims And Evidence:**

Yes

**Requested Changes:**

1. Further analysis of the FairFace dataset
2. Justification for the SiMean score

**Strengths And Weaknesses:**

## Strengths

The paper carefully investigates the claims made by B2T and offers an improved codebase, which could be valuable for researchers looking to apply the framework to their own problems. In particular, I found the following results especially interesting:

1. Quantitative evaluation of label diagnosis

The original B2T paper presented only a few cherry-picked examples for label diagnosis. In contrast, this paper provides a more rigorous, quantitative evaluation, offering a clearer understanding of the framework’s limitations.

2. Code reimplementation and workflow optimization

The authors did a commendable job reimplementing the missing components and optimizing the codebase, achieving a six-fold speedup in computation. These engineering efforts could benefit the community and improve reproducibility.

---
## Minor Weaknesses from Strengths

While the contributions are valuable, a few limitations emerged from the strengths:

1. Lack of analysis on the failure of label diagnosis

Although the paper quantitatively shows that B2T is not effective for label diagnosis, it does not investigate the underlying causes of this limitation or propose directions for improvement. Deeper analysis could offer insights for future enhancements.

2. Engineering efforts occupy significant space

The reimplementation and optimization work, though impressive, take up a substantial portion of both the Methods and Experiments sections. This somewhat distracts from the main scientific contributions and could have been presented more succinctly.

---
## Major Weaknesses

1. New results on FairFace are not carefully analyzed

The paper applies B2T to a new dataset, FairFace, and claims that it fails to capture racial biases. However, this conclusion is not fully convincing, as only the top 3 keywords are presented without showing corresponding captions or visual examples.

In particular, Table 8 shows that "black person" appears among the top 5 keywords extracted by B2T for both the "white male" and "black female" classes. This suggests that B2T may indeed capture racial biases to some extent, but a more refined keyword filtering process may be needed. Providing bias scores associated with "black person" and adjusting the threshold could better capture weaker bias signals.

2. The design of the SiMean score is not convincing

The paper proposes SiMean as an alternative to the CLIP score. It measures the relative similarity of a keyword to the image set compared to other candidate words, without distinguishing between correct and incorrect samples. This design leads to two problems:

- High similarity does not necessarily imply bias, as natural correlations (such as "red" and "firefighter") could result in high scores. The CLIP score addresses this by contrasting correct and incorrect samples.
- SiMean simply selects the top-N keywords from the candidate set. This approach fails when the true number of bias words is very small or very large, leading to poor bias identification.

---

> ### Author Response · Authors · 2025-05-14
> **Official Response by Authors**
>
> Thank you for your insightful review and detailed feedback! We have addressed your comments below and incorporated the relevant requested changes in our revision. Changes recommended by the other reviewers have been taken into account as well. The changed texts are currently blue for clarity within the revision.
>
> "Lack of analysis on the failure of label diagnosis" we have expanded a bit upon our reasoning/suspicions of the cause of this failure. We also mention that our manual searches are limited by scope and ambiguity, and thus, further research is necessary to reach a concrete conclusion.
>
> "The reimplementation and optimization work, though impressive, take up a substantial portion of both the Methods and Experiments sections"
> We acknowledge your point completely and understand that in other works, a smaller focus could be put on these parts. However, this paper is meant as a reproducibility study for the Machine Learning Reproducibility Challenge (MLRC). Therefore, we strived to emphasize the reimplementation work, as well as the optimizations done.
>
> "New results on FairFace are not carefully analyzed" - We have expanded our results to show the top five keywords with the accompanying bias scores in the results. As well as two visual examples with corresponding captions. Additionally, all extracted keywords with bias scores for the minority groups have been added to a new appendix G. Finally, an extra age-related bias is considered to gauge the generalizability of B2T even further. We hope to satisfy your point after having implemented these changes.
>
> "The design of the SiMean score is not convincing" - We agree that the SiMean score has inherent flaws that limit its ability to effectively discover biases. After careful consideration, we have moved any mention of this score to appendix E to ensure the validity of our reproduction is not compromised. In the appendix, we also emphasize the limitations of this score, whilst highlighting the fact that it does show the B2T-pipeline could theoretically work without a classifier. A direction for future works concerning no-classifier analysis, accompanied by potential starting points in recent literature, is given to finalise our contribution concerning this point. Thank you for your thoughtful description!
>
> We also made some changes according to the other reviewers. If you would like to see the specifics of this, please refer to the "Changes Since Last Submission" section. Again, we value the time and effort of writing your review and greatly appreciate your contribution, and hope we have successfully taken into account your concerns.

---

### Review · Reviewer_gyQo · 2025-05-10

**Summary Of Contributions:**

This submission reproduces and examines the findings of the paper, "Discovering and Mitigating Visual Biases through Keyword Explanation" (B2T). In the original B2T paper, the authors claim several application contributions, such as CLIP zero-shot prompting, model comparison, and label diagnosis. However, this submission finds that the label diagnosis contribution from B2T paper does not hold: B2T cannot identify pertinent labeling errors. Furthermore, this submission considers the necessity of using a classifier to perform bias analysis on new datasets, and they propose a SiMean score to substitute the CLIP score used in B2T, since calculating the CLIP score requires knowing which images are mispredicted.

**Audience:**

Yes

**Claims And Evidence:**

Yes

**Requested Changes:**

(1) Please polish Figure 2 to make sure its clarity is similar to Figure 1
(2) See weaknesses, maybe some additional applications should be provided for SiMean.

**Strengths And Weaknesses:**

Strengths

(1) This submission provides additional code to reproduce some figures in the original B2T paper.

(2) This submission proposes a new scoring system to substitute B2T's scoring system, saving some resources for training classifiers.


Weaknesses

(1) In Result 5 - Label Diagnosis, only boar and bee classes are checked, but in Table 5, Desk and Market are also listed without manually checking the results. I feel confused why these two additional classes are listed here, and I think it is not enough if the conclusion is obtained by checking only two small classes.

(2) The submission proposes the SiMean score, but no additional applications are shown using this new proposed score. Only some qualitative examples (keywords) are listed in Table 8. Therefore, it is hard to determine if this score can really replace the CLIP score used in the original B2T since B2T provides several applications.

(3) The paper writing should be improved by at least making sure all the figures are clear when zooming in.

---

> ### Author Response · Authors · 2025-05-14
> **Official Response by Authors**
>
> Thank you for your insightful review and detailed feedback! We have addressed your comments below and incorporated the relevant requested changes in our revision, as well as changes recommended by the other reviewers. The changed texts are currently blue for clarity within the revision.
>
> "In Result 5 - Label Diagnosis, only boar and bee classes are checked, but in Table 5, Desk and Market are also listed without manually checking the results. I feel confused why these two additional classes are listed here, and I think it is not enough if the conclusion is obtained by checking only two small classes."
> We agree - we've added additional explanation as to why this choice was made, and also mention that further research is necessary to reach a concrete conclusion, as our manual searches are very limited due to scope and ambiguity.
>
> "SiMean score" - After careful consideration with the other two reviews in mind, we have moved any mention of this score to appendix E. Here we also emphasize the limitations of this score, whilst highlighting the fact that it does show the B2T-pipeline could theoretically work without a classifier. As a reproducibility study for the Machine Learning Reproducibility Challenge (MLRC), our emphasis was on the reproduction and generalizability of B2T, which leaves a lower priority for a full evaluation of a new scoring system. Therefore, the appendix gives a direction for future work in line with the train of thought behind SiMean.
>
> "Please polish Figure 2 to make sure its clarity is similar to Figure 1" - Thank you for pointing this out! We have polished all figures within the paper to a higher quality.
>
> We also made some changes according to the other reviewers. If you would like to see the specifics of this, please refer to the "Changes Since Last Submission" section. Again, we truly appreciate your time and the thoughtful review you delivered.

---

### Decision · Action_Editor_WQvk · 2025-08-15

**Recommendation:** Accept as is

**Additional Comments:**

The revised manuscript addresses all reviewer concerns, including clearer label diagnosis analysis, extended FairFace results with visual examples, moving the SiMean score to the appendix with noted limitations, and improved figure clarity. Thus, the AC recommends Accept as is.

**Audience:**

Yes

**Audience Explanation:**

The paper tackles bias detection and mitigation in vision-language models, a topic with broad appeal in the ML community. Its clear assessment of which parts of B2T hold and which do not offers practical insights for researchers in fairness, reproducibility, and bias analysis.

**Claims And Evidence:**

Yes

**Claims Explanation:**

The study uses the original code and re-implementations to systematically test each claim, clearly showing which hold and which do not. Results are supported by quantitative evaluation and extended experiments, and limitations of less-validated parts (e.g., SiMean) are openly acknowledged.